



# The 4.2 ka BP Event in the Mediterranean Region: an overview

Monica Bini[1], Giovanni Zanchetta[1], Aurel Perșoiu[2], Rosine Cartier[3], Albert Català[4], Isabel Cacho[4], Jonathan R. Dean[5], Federico Di Rita[6], Russel N. Drysdale[7], Martin Finnè[8], Ilaria Isola[9], Bassem Jalali[10], Fabrizio Lirer[11], Donatella Magri[6], Alessia Masi[6], Leszek Marks[12], Anna Maria Mercuri[13], Odile Peyron[14], Laura Sadori[6], Marie-Alexandrine Sicre[10], Fabian Welc[15], Christoph Zielhofer[16], and Elodie Brisset[17]

[1]Dipartimento di Scienze della Terra, University of Pisa
[2]Emil Racovita Institute of Speleology, Romanian Academy Transylvania, Romania
[3]Postdoctoral researcher at Lund University Quaternary Sciences, Sweden
[4]GRC Geociències Marines, Departament de Dinàmica de la Terra i de l'Oceà, Facultat de Geologia, Universitat de Barcelona, Spain
[5]School of Environmental Sciences, University of Hull, Hull, UK
[6]Dipartimento di Biologia Ambientale, University of Rome "La Sapienza", Rome, Italy
[7]School of Geography, University of Melbourne, Australia
[8]Department of Archaeology and Ancient History, Uppsala University, Uppsala, Sweden and Department of Physical Geography, Stockholm University, Stockholm, Sweden
[9]Istituto Nazionale di Geofisica e Vulcanologia, Sezione di Pisa, Pisa, Italy
[10]LOCEAN Laboratory, Sorbonne Universités (UPMC, Universitè de Paris 06)-CNRS-IRD-MNHN, France
[11]stituto per l'Ambiente Marino Costiero–CNR, Naples, Italy
[12]Faculty of Geology, University of Warsaw, Warsaw, Poland
[13]Dipartimento di Scienze della Vita, Università di Reggio Emilia e Modena, Italy
[14]Institut des Sciences de l'Evolution (ISEM), Université de Montpellier, France
[15]Institute of Archaeology Cardinal Stefan Wyszynski University in Warsaw, Warsaw, Poland
[16]Chair of Physical Geography, Leipzig University, Leipzig, Germany
[17]IPHES, Institut Català de Paleoecologia Humana i Evolució Social, Tarragona, Spain and Àrea de Prehistòria, Universitat Rovira i Virgili, Tarragona, Spain

**Correspondence:** Monica Bini (monica.bini@unipi.it)

**Abstract.** The Mediterranean region and the Levant have returned some of the clearest evidence of a climatically dry period occurring around 4200 years ago. However, some regional evidence are controversial and contradictory, and issues remain regarding timing, progression and regional articulation of this event. In this paper we review the evidence from selected proxies (sea-surface temperature, precipitation and temperature reconstructed from pollen, $\delta^{18}$O on speleothems, and $\delta^{18}$O on lacustrine

5   carbonate) over the Mediterranean basin to infer possible regional climate patterns during the interval between 4.3 and 3.8 cal ka BP. The values and limitations of these proxies are discussed, and their potential for furnishing information on seasonality is also explored. Despite the chronological uncertainties, which are the main limitations for disentangling details of the climatic conditions, the data suggests that winter over the Mediterranean was drier condition, in addition to already dry summers. However, some exceptions to this prevail, - where wetter condition seems to have persisted - suggesting regional heterogeneity

10   in climate patterns. Temperature data, even if sparse, also suggest a cooling anomaly, even if this is not uniform. The most





common paradigm to interpret the precipitation regime in the Mediterranean – a North Atlantic Oscillation-like pattern – is not completely satisfactory to interpret the selected data.

*Copyright statement.* TEXT

# 1 Introduction

In recent years, it has become paradigmatic that the Holocene was a relatively stable climatic epoch when compared to the last glacial period (e.g., Dansgaard et al., 1993). However, long-term, astronomically driven changes in insolation produced changes in temperature (Marcott et al., 2013), but see also Marsicek et al. (2018), associated with a progressive southward shift of the Intertropical Convergence Zone (ITCZ) and a weakening of Northern Hemisphere summer monsoon systems (e.g., Wright et al., 1993; Fleitmann et al., 2003; Braconnot et al., 2007). A number of short, multidecadal-to-centennial-scale climatic events, the origin of which often remains unclear, is superimposed over this long-term trend (e.g., Denton, and Karlén, 1973; Bond et al., 1997; ; Wanner et al., 2011). At the regional-to-global scale, some events appear synchronous and linked to specific changes in circulation patterns (e.g., Trouet et al., 2009; Dermody et al., 2012; Zanchetta et al., 2014). A good example is the Medieval Climate Anomaly in the Atlantic region, which has been explained in terms of an anomalously persistent positive mode of the North Atlantic Oscillation (NAO) (Trouet et al., 2009). However, the synchronicity and therefore the origin of many such events remains challenging. A major and much-discussed example of a multidecadal-to-century-scale event is the so-called "4.2 cal. ka BP event". The detection of this event over an extensive region, and its common expression as an interval of cooling and drying (e.g., Cullen et al., 2000; Drysdale et al., 2006; Dixit et al., 2014), points to a global "megadrought" (Weiss, 2015, 2016). The significance of the climate event 4.2 ka cal BP at global scale has been accepted recently as the formal boundary of Late and Middle Holocene at 4250 yr b2k (www.stratigraphy.org). Despite its near-pervasive recognition, the timing, duration and progression of this event have yet to be defined in detail, whilst its origin in terms of changes in ocean and atmospheric circulation remains elusive (Booth et al., 2005; Zanchetta et al., 2016; Carter et al., 2018). Moreover, not all the palaeoclimate records preserve evidence of the 4.2 cal ka BP event, at least as a prominent feature of the late Holocene(e.g., Seppa et al., 2009; Göktürk et al., 2011; Roland et al., 2014) and not necessarily as a cold and dry event (e.g., Railsback et al., 2018). Some researchers have suggested that this event is best described as a complex succession of dry/wet events, rather than a long, single dry event (Magny et al., 2009; Railsback et al., 2018), further complicating the matter. The Mediterranean region shows some of the most consistent evidence of the 4.2 cal ka BP event. It is mostly recognized as a dry interval, and is identified in pollen records (e.g., Magri and Parra, 2002; Di Rita and Magri, 2009; Kaniewski et al., 2013), speleothems (Drysdale et al., 2006; Cheng et al., 2015; Zanchetta et al., 2016; Finné et al., 2017), lakes (e.g., Zanchetta et al., 2012b) and marine sediments (e.g., Margaritelli et al., 2016). However, the chronology of the event is not precisely defined and, in many records, the event is not evident (Finné et al., 2011), challenging the view of a generalized period of significant drought. In this paper, we review the evidence, nature and chronology of the 4.2 ka BP event in the Mediterranean region by comparing different marine and





terrestrial proxy records. This will serve to identify gaps in the regional coverage, to expose aspects that should be addressed in future research on this topic, to determine if coherent regional/subregional climatic patterns are present, what their links are to regions further afield, and if such patterns can be plausibly explained in a coherent meteo-climatic framework.

## 2 Methods and terminology

In this paper we use the term "4.2 cal ka BP event" to indicate a period of time between ca. 4.3 and 3.8 cal ka BP (close to the definition of Weiss (2015, 2016), whilst being mindful that this does not necessarily correspond to the true temporal evolution of the climatic event, but the chronological interval where often this event is recognized. We have considered a large set of records for this review. In the end, the records selected for inclusion are those possessing robust age models and high-resolution time series (i.e. at least sub-centennial). It has been recognized that chronology for some Mediterranean records could be problematic, as demonstrated, for instance, using tephra layers as chronological points (Zanchetta et al., 2011, 2016, 2018). However, in the absence of these chronological control points, the question of exclusion or inclusion of records involves a degree of subjectivity. For example, we argue that only records dated by radiocarbon using terrestrial remains should be selected. Marine records dated using radiocarbon on foraminifera can show millennial-scale change of the reservoir effect (Siani et al., 2001), and different degrees of bioturbation, which can complicate comparisons between different archives. Some speleothem records, dated in the past with U/Th methods, have chronologies inconsistent with more recent accurate age determinations (e.g., Grotta di Ernesto McDermott et al., 1999; Scholz et al., 2012). However, to have a wide regional coverage with proxy records we have also included records with relatively low resolution and with age control that is not necessarily optimal. With this in mind, we are also aware that our selection of records could appear incomplete for some archives/proxies. For each archive, we have selected the most powerful (or considered as such) proxy to reconstruct climate. It is obvious that many archives are suitable for a multiproxy approach, but some proxies can be more related to local processes, and the correlation with climatic variables less direct than others can. Moreover, it would be useful to use similar proxies in different environments, even if they do not necessarily have the same meaning (Roberts et al., 2010). We must also consider that the scale and longevity of human activity around the Mediterranean may create locally serious difficulties in distinguishing climate change from human impact (e.g. deforestation, erosion) in many proxy records of past environmental change (England et al., 2008; Roberts et al., 2004, 2010). The 4.2 cal ka BP event in the Mediterranean (including the Levant) is strictly related to complex societal evolution and development at the basin scale (Weiss, 1993; Zanchetta et al., 2013), and care is necessary in interpreting proxy records where local factors override regional climate changes. Pollen records are surely one of the most important sources of information on past environment in the Mediterranean and they will be used in this review, but they are one of the proxies that has been suggested to be seriously compromised by human activity (e.g. Roberts et al., 2004; Fyfe et al., 2015, 2018). Given the importance of having estimates of past temperature and precipitation reconstruction, we have selected pollen-based quantitative reconstructions (e.g. Peyron et al., 2017). In terrestrial archives, in addition to pollen data, we selected the oxygen isotope composition of lacustrine carbonates and speleothems as the main proxies of past climate due to their potential for preserving strong hydrological signals (Bar-Matthews et al., 1996; Roberts et al., 2008, 2010). For marine





records, sea-surface temperature (SST) reconstruction was preferred to oxygen isotope composition of planktonic foraminifera, for the unavoidable limitation of the latter to represent the mixing signal of temperature and changes in local seawater isotopic composition (i.e. salinity). These are the main proxies considered for our reconstruction: they show the largest coverage and the most complete, in our opinion, climate information. These proxies also permit, to some extent, the disentanglement of climate

signals between the cooler and warmer seasons, as we will propose. We are aware that there are limitations in this, but it is necessary to understand more details about the 4.2 cal ka BP event. There are other proxies which can give potentially further important information, like lake-level changes (Magny et al., 2007, 2011), and although discontinuous, isotopes on paleosols (Zanchetta et al., 2000, 2017) or paleofloods (Zielhofer and Faust, 2008). Although rare, dust records appear of particular relevance in informing about past circulation patterns and hydrological conditions (e.g., Zielhofer et al., 2017b). However,

these records have still low regional coverage and will only be referred to briefly in the discussion.

## 2.1 Selected archives and proxies

Table 1 and Fig. 1 show the complete list of selected records, including the original references and the proxy considered.

### 2.1.1 Speleothems

The number of speleothem records covering the Holocene with appropriate resolution has dramatically increased in recent

years although they are geographically unevenly distributed (e.g., McDermott et al., 2011; Deininger et al., 2017). Multiple proxies obtained from speleothem calcite are often interpreted as hydrological indicators and, in particular, the oxygen isotope composition ($\delta^{18}$O) is the most common proxy utilized (Lachniet , 2009). In the Mediterranean basin, in many instances the $\delta^{18}$O records are seen as an indicator of the amount of precipitation recharging the cave (the so-called "amount effect", Bar-Matthews et al., 1996; Bard et al., 2002; Zanchetta et al., 2014; Finné et al., 2017), with higher (lower) $\delta^{18}$O values of calcite

indicating drier (wetter) conditions. This is true only when considered in terms of long-term changes in the isotopic composition of seawater sources of the precipitation ("source effect" e.g., Cheng et al., 2015). The interpretation of the $\delta^{18}$O record as an indicator of hydrological changes is supported, in some instances, by other proxies like trace elements (Drysdale et al., 2006; Regattieri et al., 2014; Wassenburg et al., 2016) which should be more common in the future. More refined interpretations indicate that the $\delta^{18}$O composition of cave recharge is in some cases related to North Atlantic Oscillation (NAO) (Smith et al.,

2016; Wassenburg et al., 2016), even if investigations of $\delta^{18}$O composition of precipitation do not always reproduce a fidelity with NAO pressure patterns (Field , 2010; Baldini et al., 2008). Domínguez-Villar et al. (2017) suggest that most of the isotopic signal in Iberian speleothems is not principally related to the amount of precipitation but rather changes in the ratio of recycled precipitation, and is correlated to pressure patterns over the North Atlantic. Therefore, some authors suggest that the $\delta^{18}$O of the speleothem calcite is a direct expression of the North Atlantic influence and state, especially in the western Mediterranean

(e.g., Smith et al., 2016; Wassenburg et al., 2016) during winter months. However, a change in provenance of precipitation (Holmes , 2010) and precipitation amount during winter and summer months can further complicate the interpretation of the $\delta^{18}$O records. Therefore, a unifying and completely satisfactory explanation for $\delta^{18}$O of calcite is probably not yet available throughout the Mediterranean area and surrounding regions (Moreno et al., 2014). It is important also to remember that the



$\delta^{18}$O signal is skewed towards the period of calcite precipitation and its relation with cave recharge. As anticipated, authors tend to assume that most of the cave recharge occurs during winter (or autumn-winter) and most of the $\delta^{18}$O signal should be related to this condition (Deininger et al., 2017). In some instances, the complex interpretation of $\delta^{18}$O as a direct climatic proxy has led different authors to prefer $\delta^{13}$C of speleothem calcite as better hydrological indicator of local conditions (e.g., Genty et al.,

2006; Göktürk et al., 2011). The number of factors influencing the final $\delta^{13}$C value of a speleothem (e.g., Mühlinghaus et al., 2009) make this proxy probably just as, if not more, complicated as $\delta^{18}$O, and in addition, for the strong influence of soil-$CO_2$ production on the final $^{13}$C/$^{12}$C ratio on speleothems, a change in land use and deforestation can have a particularly prominent effect making it particularly sensitive to human impact above and within the cave. It is usually reported that speleothems possess a superior chronology compared to other archives thanks to the U/Th technique (e.g., Richards and Dorale, 2003).

However, this assumption is strictly true for speleothems acting as closed system for the U (Bajo et al., 2016), and with only minor clastic contamination (Hellstrom , 2006). For this review, we have selected 16 records (Fig. 2; Table 1). The main reason for excluding some records is the presence of long hiatuses (thousands of years) over the 4.2 cal ka BP event that may not necessarily relate to climatic conditions (e.g.Villars, Chauvet and La Mine caves, Genty et al., 2006), (Carburangeli cave Frisia et al., 2006). Some records have been rejected for their U/Th chronologies, as shown in later studies (e.g. Savi Cave, Frisia et

al., 2005), ages disputed in Belli et al. (2013). However, shorter hiatuses coherent with isotopic changes are considered here as evidence of particularly dry and potentially cooler climate conditions (i.e. Trypa cave, Finné et al., 2017). In this regard, Stoll et al. (2013) interpreted the growth cessation of many speleothems at ca. 4.1 ka in NW Spain to be caused by increased aridity since this time. However, this is a general signal and not specifically related to a short interval, suggesting that eventual increasing in aridity during the 4.2 cal ka BP event is within a general frame of increasing aridification.

### 2.1.2  Lacustrine settings

The oxygen isotope composition of lacustrine carbonates in the Mediterranean region is usually interpreted as mainly being controlled by changes in the isotopic composition of lake water (Roberts et al., 2008), which is controlled by different factors, including changes in the isotopic composition of precipitation and the degree of evaporation (Zanchetta et al., 2007b; Leng et al., 2010a). Each lacustrine setting has a different set of responses and different water $^{18}$O-enrichement (Roberts et al., 2008;

Leng et al., 2010a, b) due to evaporative effects, which depend on several factors, namely temperature, relative humidity, wind fetch and strength, residence time of body water (e.g., Craig et al., 1965). Different types of carbonates (e.g. ostracods, freshwater shells, bio-induced carbonates) may precipitate during different parts of the year, with bio-induced calcite (endogenic) often related to spring to summer algal bloom (Leng and Marshall, 2004). Therefore, the $\delta^{18}$O of endogenic carbonate will tend to be weighted towards the summer conditions, although care is necessary for their interpretation and more complex options

have been proposed (e.g., Zielhofer et al., this issue). Endogenic carbonates can be contaminated by clastic carbonates (Leng et al., 2010b) and/or early diagenetic minerals (i.e. siderite, Lacey et al., 2016), which can degrade the paleoclimate signal and must be carefully evaluated case by case. Despite these possible complications, trends toward higher (lower) $\delta^{18}$O values are generally explained as an indication of drier (wetter) conditions (Zanchetta et al., 1999, 2007b; Roberts et al., 2008; Leng et al., 2010a, 2013). Reduction in general lake recharge and particularly arid conditions during the warmer part of the year

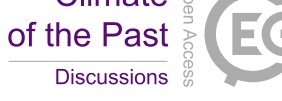

favor higher $\delta^{18}$O values of endogenic carbonate. As is the case of speleothem records, lacustrine $\delta^{18}$O records are unevenly distributed over the Mediterranean basin (Fig. 1, Table 1) and this represents an important limitation to regional interpretations. Following the list proposed in the important review of Roberts et al. (2008), very few new lacustrine records have been added since e.g.Lake Ohrid (Lacey et al., 2015), Lake Prespa (Leng et al., 2010a, 2013), Lake Yammoûneh (Develle et al., 2010),

Lake Shokdra (Zanchetta et al., 2012b), Sidi-Ali (Zielhofer et al., 2017a). Some of the records reported in Roberts et al. (2008) and some new records were too short, with too low resolution and poor chronologic accuracy (e.g. Lake Pergusa, (Zanchetta et al., 2007b), Valle di Castiglione, (Zanchetta et al., 1999), Lake Yammoûneh (Develle et al., 2010)) and have been excluded from this review. We have to note that some records (particularly in the past) used different kinds of organic matter and carbon (i.e. shells) for radiocarbon dating (e.g. , Baroni et al., 2006), which can be affected by different reservoir and hardwater effects

of unknown amount, leading to significant offsets between records. Fig. 3 shows the selected lacustrine records.

### 2.1.3   Marine records

For marine records we selected sea-surface temperature (SST). Figure 4 shows a compilation of 12 published sea surface temperature (SST) records from coastal and deep-sea sites of the Mediterranean Sea. Apart from Figure 4b, which is based on the Mg/Ca ratios in planktonic foraminifera *Globigerina bulloides*, all the records are based on alkenone paleothermometry.

As for other archives, the comparison of multiple proxies and site compilation require consideration of potential seasonal biases (Emile-Geay et al., 2017). Maximum production of alkenones in the Ligurian, Alboran and Adriatic seas would take place mainly in spring and autumn (Ternois et al., 1997; Totti et al., 2000; d'Ortenzio and Ribera d'Alcalà, 2009), while in other sub-basins, such as the Balearic Sea and the Bannock Basin, primary production exhibits a less clear pattern, with maximum algal blooms during spring (d'Ortenzio and Ribera d'Alcalà, 2009; Ziveri et al., 2000). However, several high-resolution

alkenone-derived SST records, which overlap the post-industrial period and allow comparison with SST observations, highlight a consistent match between alkenone-SST and average annual sea-surface temperature (Sicre et al., 2016; Jalali et al., 2018; Nieto-Moreno et al., 2013; Moreno et al., 2012; Cisneros et al., 2016). The single Mg/Ca SST record from the Alboran Sea is mainly believed to reflect spring SSTs (Jiménez-Amat and Zahn, 2015). Cisneros et al. (2016) observe that modern regional oceanographic data indicate that *Globigerina bulloides* Mg/Ca is mainly controlled by SST of April-May related to the

primary bloom productivity.

### 2.1.4   Pollen data

The selection of pollen data is probably the more complex considering the elevated number of the sedimentary successions analyzed over the basin and in different settings (marine and lacustrine cores). Using only records with acceptable resolution (i.e. resolution chronologically higher than interval considered: Peyron et al., 2017), with the reconstruction of precipitation

and temperature, the number of records is, however, strongly reduced. The basic assumption in the pollen-based climate reconstructions (assemblage approach or transfer function) is that modern-day observations and relationships can be used as a model for past conditions and that the pollen-climate relationships have not changed with time (Birks, 2005). Among the main approaches available to reconstruct quantitatively past climate from pollen data, most of the selected records have been performed

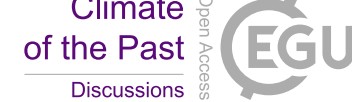

using the Modern Analogue Technique (MAT) (Guiot , 1990), an "assemblage approach" frequently used in climate reconstructions. This method was successfully used for the Holocene climate reconstructions from terrestrial and marine records (e.g. , Peyron et al., 2011, 2017; Mauri et al., 2014, 2015). The MAT is based on a comparison of past assemblages to modern pollen assemblages. An important requirement is the need of a high-quality training set of modern samples. The training set should be

representative of the likely range of variables, of highest possible taxonomic detail, of comparable quality and from the same sedimentary environment (Brewer et al., 2013). It must cover a wide environmental range. Limitations with using MAT are the occurrence of no analogues or multiple analogues (Birks et al., 2010), and the potential problem of the spatial autocorrelation with the MAT but also in the transfer functions (Telford and Birks, 2005). All these interpretations hold if minor human impact is considered in vegetation and pollen production, which may be a major concern for some reconstructions (Fyfe et al., 2015,

2018). Figure 5 shows the reconstructed temperatures (annual) and precipitation (annual, winter, and summer) through the 4.2 ka cal BP event.

## 3  Discussion

### 3.1  Chronology: the Achilles heel of the problem

The first general observation is that for many of the records, chronological uncertainties and different, and sometimes poor,
chronological resolution are the main obstacles in the precise identification of the event, its timing, duration and progression. At this stage, both aspects seem to be an unavoidable limitation for an in-depth understanding of this interval. For example, we show selected pollen records (Arboreal Pollen, AP%) containing the Avellino tephra plotted with their published age model (Fig. 6) in the Central Mediterranean. Quite apparent are the century-scale differences in age models presented, considering the well-constrained ages of this tephra layer (ca. 3.8 cal. ka BP: see discussion and references in Zanchetta et al. (2018).
It is reasonable to assume that similar levels of uncertainties may be present in other records. We note that in Figure 6, the identification of the 4.2 cal. ka BP appears problematic (at least using AP% signal), which is not in the case for other pollen records in Italy (Magri and Parra, 2002; Di Rita and Magri, 2009; Di Rita et al., 2018a) or in the Levant (e.g., Kaniewski et al., 2013; Kaniewski et al. this isuue, 2018). Interestingly, some records may suggest that the 4.2 cal. ka BP interval is characterized by several important oscillations rather than one simple long interval of specific (usually drier) climatic conditions (see for
instance Fig. 2 Skala Marion and Solufar or GLD1, or in SST in the Gulf of Lion and Alboran Sea in Fig. 4), as it has been suggested for other parts of the world (Railsback et al., 2018) and in this issue (Kaniewski et al.). However, there is no clear and coherent evidence of this in many of the different selected records. A good example of the complexity of the evidence and the link to chronological accuracy is the lake level record at Accesa. This was used as the archetypal example to demonstrate that, in reality, the event is "tripartite" (Magny et al., 2009), where a phase characterised by drier conditions at ca. 4100-
3950 cal. BP appeared bracketed by two phases marked by wetter conditions and dated to ca. 4300-4100 and 3950-3850 cal. BP, respectively (Magny et al., 2009). Magny et al. (2009) reported a significant number of records over the Mediterranean showing reasonably the same climatic evidence. However, subsequent works using tephra layers (Avellino and other tephras are present in Lake Accesa: Magny et al., 2007) showed the inconsistency of this detailed correlation and the 4.2 cal ka BP



event should be dominated by lower lake level (Zanchetta et al., 2012a, b, 2016), supporting the existence of a prominent drier phase. To circumvent some of the complex issues related to chronological problems, authors have used two different approaches. The first, and the most common, is an accurate selection of records which show conspicuous and chronologically consistent evidence of the event (e.g., Drysdale et al., 2006; Magny et al., 2009; Kaniewski et al. this isuue, 2018). Records

in which the expression of the event is equivocal are usually removed. A different approach is more "climatostratigraphic", which is used instead of a simple chronological selection of a time window to correlate the event on the basis of similarity of the climatic curve. This has the obvious limitation that any regional articulation and/or timing progression will be lost. For instance, we can force the correlation of Skala Marion $\delta^{18}$O record with those of Renella, Mavri Trypa and Solufar caves (Fig. 7 for instance, Psomiadis et al., 2018) implicitly followed this approach in their fig. 6, for Skala Marion, Renella and Solufar

cave records) to assume that the interval at ca. 3.9-3.4 ka characterized by higher $\delta^{18}$O values at Skala Marion correspond to similar interval at ca. 4.3-3.8 ka well identified in Renella and Mavri Trypa. Therefore, we are aware that the chronological issue can heavily contaminate the following discussion. In the following sections, we separate proxies on the basis of their presumed meaning: annual average vs seasonal component. Maps are produced (Fig. 8) specifying the local record condition (i.e. warmer/cooler and/or drier/wetter). For reasons that may depend on chronology, proxy sensitivity and/or local response

to climatic change, it is not always obvious to define the specific environmental conditions of the considered interval. In some instances, changes lasted longer than the interval considered, or during the interval there is a clear change in conditions, or the interval is characterized by invariant conditions and/or by trends. Once again, some margin of subjectivity may have existed in the evaluation of a single record. Generally, for each site, if most of the interval is dominated by specific conditions (wetter, drier, colder, and/or warmer), this is represented in the maps accordingly. If the environmental trend moves toward a specific

state, the site is defined by this trend (e.g. if the trend during the interval is toward drier conditions, the site is deemed "drier" during the event). Ambiguities are still possible and are indicated where appropriate.

### 3.2    The annual average conditions

Based on SST and pollen reconstructions, it is possible to gain some insights on the average conditions during the 4.2 cal ka BP interval (Fig. 8a and 8b). As can be seen from Figure 4, SST records from 8000 to 2000 yr BP show strong differences

in their temporal resolution. Some poorly resolve the 4.2 ka event (e.g. Fig. 4e,f). In the Alboran Sea cores, MD95-2043 and ODP-976 show substantially invariant alkenone temperatures, despite their rather high-resolution, whereas in the Mg/Ca SSTs record from the Alboran Sea (ODP-976, Fig. 4b; Jiménez-Amat and Zahn, 2015) documents a cooling during spring rather than the mean annual conditions. The alkenone-SSTs from the Gulf of Lyon (Jalali et al., 2016) (Fig. 4c) indicate several SST oscillations with some important cooling in the final part of the interval. BS79-38 in the Tyrrhenian Sea shows essentially

invariant SSTs, whereas in the Ionian sea core M25/Kl11 and M40/4-SL78 (Fig. 4f), despite their low resolution, show an opposite trend. In the Adriatic Sea, an apparent modest warming is present only in core AD91-17 (Giunta et al., 2001). In the Levantine Basin, an apparent general tendency for cooling seems to be present in core GeoB7702-3 (Fig. 4h) (Castaneda et al., 2010), after a phase of warming even if the interval considered comprises the descending part of a longer SST trend. In the Nile prodelta area, a warmer interval seems to be present in the core MD04-2726 (Fig. 4i) (Jalali et al., 2017), even if with some



oscillations. Figure 5 shows reconstructed annual average temperatures in different parts of the basin using pollen records. Most of the central Mediterranean records (except Trifoglietti, which shows some intermediate behavior, with tendency of warming) shows a cooling at this time. Lake Malik shows a long-term cooling, rather a precise interval of cooling. In contrast, in the Akko/Acre record there is first a period of warmer conditions followed by a later period with a prominent cooling, and

this trend, according to Kaniewski et al., (this issue), is consistent with other sites in the Eastern Mediterranean. Figure 8a shows that the few records selected show generally lower average temperatures in the Western-Central Mediterranean, whereas towards the eastern part higher temperatures seem to prevail. Annual precipitation estimated from pollen show several records with clear evidence of reduced precipitation (Malik, Pergusa, Trifoglietti, Akko/Acre), even if this signal is generally complex (Fig. 5), with part of the selected records not suggesting drier conditions (e.g. Malta, SL152, MD95-2043). Indeed, figure 8b

highlights that poor data coverage for estimating past annual precipitation prevents any detailed considerations.

## 3.3   Winter records

In this reconstruction, we have included speleothems and pollen data, whilst being aware of the limitations discussed in sections 2.1.1 and 2.1.4. Most of the records indicate drier conditions in winter during the 4.2 cal ka BP event. Speleothems are the most conspicuous record. Qualitatively, six out 15 records show that during the considered interval there is a clear increase in

$\delta^{18}O$ values (Fig. 2; Jeita, Ascusa, Poleva, Mavri Thrypa, Renella, and Gueldaman caves). To this group can be added Soreq and Corchia (stalagmite CC26) showing a modest increase of values at that time. Two caves (Grotte de Piste and Keita) show a clear interval of decreasing $\delta^{18}O$ values (Fig. 2). Sofular and Skala Marion show a similar pattern of oscillating behavior, with an important part of the period characterized by a marked decrease in the $\delta^{18}O$ values. A similarly, but not identical, pattern is present in the Iberian caves, Ejulve and Cueva de Asiul. Invariance defines the Ernesto record. This apparently contradictory

behavior is regionally well defined with the Central Mediterranean (Algeria, Central Italy, Romania, and Western Greece) and part of the Levant, characterized by the most marked and significant interval characterized by higher $\delta^{18}O$ values. Instead, North Iberian Peninsula and North Morocco show a tendency to lower or oscillating values, as does the opposite end of the basin with the records from Sofular and Skala Marion caves. Accepting the fact that $\delta^{18}O$ represents the amount of precipitation during the winter recharge period, we observe that the central part of the basin, with some extension to the Balkans and Romania,

shows decisively drier conditions compared to the other analyzed sectors, where they show wetter or at least more variable conditions. Winter precipitation reconstructed from pollen show clear indications of decreased precipitation at Ledro, Pergusa, MD90-917, and MD95-2043 (even if modest). Drier conditions are evident for most of the interval at Trifoglietti (with a late recovery toward wetter conditions), whereas at Malta most of the interval shows wetter conditions compared to the previous period. Accesa shows a strong oscillatory behavior, with the central part of the drier interval bracketed by two wetter intervals.

However, the general trend of Accesa cannot be classified unambiguously as wetter or drier. Figure 8c shows the prevailing precipitation conditions (wetter/drier for the interval considered) during winter for the proxy in question. Geographically, it seems quite consistent that most of the Mediterranean records show drier conditions during winter. Despite the poor coverage, the possible exceptions are Morocco and the Iberian Peninsula, as well as some sectors of the Eastern Mediterranean, possibly indicating a regional articulation.



### 3.4 Summer conditions

As discussed in section 2.1.2, endogenic lacustrine carbonates can be considered reasonably as a first-order hydrological (precipitation minus evaporation) signal of summer, even if influenced by the effect of recharge during previous periods. Some records (Sidi−Ali, Ohrid, Hula) show no peculiar trends during the period considered, although for a very short interval

centered at ca. 4.2 cal ka BP (Zielhofer et al. this issue) observe a minor oscillation in $\delta^{18}$O values interpreted as increased winter recharge (Fig. 3). Other records show a clear peak towards more positive values (Medina, Frassino, Shkodra, Prespa, Dojran, Nar, Golhisar), whereas others show a tendency to decrease values (e.g Van) or a clear peak of lower $\delta^{18}$O values (i.e. Ioannina). Instead, Zeribar and Mirabad records show a trend towards lower $\delta^{18}$O values, even if the resolution of the two records for this interval is rather poor. A number of records show a well-marked peak of increasing $\delta^{18}$O values within

the considered interval but show a different duration, possibly due to differences in age models and resolution. It is surprising to note that the two-sister lakes Prespa and Ohrid (Lacey et al., 2015; Leng et al., 2010a, b, 2013), exhibit, for this interval, significantly different trends (Fig. 3). This can be explained by the effect of higher residence time and more dampened isotopic composition of the lake water at Lake Ohrid due to its large recharge by karst springs (more than 50%, Wagner et al., 2017) in comparison to Lake Prespa (Leng et al., 2010a). This renders the latter much more sensitive to hydrological changes, as

demonstrated by its dramatic lowering of lake−level changes in recent years (van der Schriek and Giannakopoulos, 2017). Regionally, and consistent with Lake Prespa, are the data from Lake Dojran and Shkodra (Fig. 3) which show drier conditions. Interestingly, the Ioannina record shows a very marked phase of lower $\delta^{18}$O values in an almost perfect antiphase with the other nearby lakes. We note that Ioannina is a record obtained using ostracods; instead, the others were obtained by measuring the isotopic composition of endogenic calcite, which may integrate the isotopic signal of a number of years. Possibly, the Ioannina

record intercepts a period of particularly pronounced snow melting in spring, while the other lakes record a significantly longer period of evaporated waters during spring/summer. Progressive trends towards lower values shown by Zeribad and Mirabad are difficult to interpret because of their relatively low resolution, although there is regional coherence. Some pollen reconstructions show evidence of drier conditions (Fig. 5; Ledro, Trifoglietti, SL152−PA, and MD95−2043, even if the last is within a longer period of summer-reduced precipitations), but others indicate a tendency towards increasing precipitation and/or decisively

wetter than the previous interval (Fig. 5, Malik, Accesa, Pergusa, and MD95-2043). Figure 8d shows the regional pattern of the considered summer conditions. Despite the large gaps in record coverage, once again, most of the records indicate drier conditions, even if in the central sector (Italian peninsula and possibly some sectors of Greece) and in the eastern end, a tendency toward wetter condition may exist.

### 3.5 Is a synthesis possible?

The Mediterranean basin is located in a transitional zone between North Africa and Arabian arid regions, dominated by the subtropical high-pressure system, and central and Northern Europe where mid-latitude westerly circulation dominates. The basin is also exposed to the indirect effect of the Asian and African Monsoons in summer and to western Russian/Siberian High systems in winter (e.g. Lionello et al., 2006, and reference therein). Therefore, to look for a simple mechanism for explaining



the 4.2 cal ka BP event is not a simple task. The uneven distribution of many proxy records, and the previously discussed concerns on chronology, can make general conclusions and detailed regional articulation difficult. It is beyond the scope of this contribution to discuss the detailed mechanism and forcing, however, considering the discussion made on previous sections regarding the limitations of proxies and our approach in their interpretation, some interesting points emerge. Based on Figure

8a the average annual temperature seems to show a tendency of cooling for most of the basin. Even when moving from West to East there seems to be an increase in the number of records showing an increase in temperature, instead of cooling, suggesting a possible coherent trend. Data for average annual precipitation are sparse and make syntheses difficult (Fig. 8b). Figure 8c shows the situation regarding winter precipitation inferred from speleothem $\delta^{18}$O records and pollen reconstructions. They are of particular relevance for most of the basin, which is strongly controlled by NAO, in particular in the western and central part

of the basin (Lionello et al., 2006; López-Moreno et al., 2011), with winter precipitation being negatively correlated with NAO. On the contrary, areas of the southeastern Mediterranean show an anti-correlation with western Mediterranean precipitation, resulting in a see-saw pattern known as the Mediterranean Oscillation (MO, e.g. Conte et al., 1989). This clearly opens the possibility that seemingly complex patterns in precipitation are not just an artefact of the proxies and/or chronology, but can be a real and robust climatic pattern. On the other hand, the present climatic configuration can have different past regional

expressions due to a combination of multiple factors. Indeed, antiphasing between different sectors of the Mediterranean basin has been found during the Middle Age Climate Anomaly and the Little Ice Age between Iberian Pensinsula and Turkey, with the latter which is not the present day center of action of the MO (Roberts et al., 2012). This has also been suggested during several late Holocene oscillations including the 4.2 ka event, in antiphase in the south-western and the south-central Mediterranean regions (Di Rita et al., 2018b). The pattern described in fig. 8c, with the main distribution indicating pervasive

drier conditions over most of the Mediterranean during winter, is consistent as a NAO-like positive mode, where westerly sourced vapor masses shift northward owing to a pronounced Azores high. NAO positive mode during this period is also supported by the concentration of TERR-alkanes in the Gulf of Lyon shelf sediment (Jalali et al., 2016). However, the presence of a pole of possibly wetter conditions over Morocco (as indicated by La Piste cave, Fig. 6 and 8c, and partly also by Sidi-Ali lake) and the northern Iberian Peninsula, and a less evident and questionable wetter area in the eastern sector (Solufar and

Skala Marion caves, fig. 2), suggest a different configuration. Indeed, at La Piste Cave, periods characterized by lower $\delta^{18}$O values have been interpreted as a period of negative NAO-like conditions (Wassenburg et al., 2016), which is also suggested by other authors for this period (e.g. Di Rita et al., 2018a). However, this interpretation runs counter to the speleothem evidence from central Mediterranean indicating drier conditions (Renella, Corchia, Mavri Trypa, Drysdale et al., 2006; Regattieri et al., 2014; Finné et al., 2017). Figure 9 shows the NAO index inferred by Olsen et al. (2011); during the 4.2 cal ka BP event,

the NAO is mostly positive, if not particularly prominent, suggesting that NAO configuration alone would not be particularly useful to interpret this period of time. Moreover, figure 10 shows that a negative NAO-like configuration with a similar pattern of precipitation like today is unlikely considering the data of figure 8c. It is interesting to note that the distribution of Figures 8c and 10 seems to find some similarity with the reconstruction proposed by Dermody et al. (2012) during Roman Time. According to Dermody et al. (2012) the dominant pattern of variability in humidity between 3000-1000 yr BP shows a seesaw

with Spain and Israel in one side and the Central Mediterranean on the other. The pattern in climatic humidity are similar





to precipitation anomalies associated with the East Atlantic/West Russia pattern which today represents a secondary mode of precipitation pattern during winter within the dominance of the North Atlantic Oscillation (NAO) pattern (Xoplaki et al., 2004). It is clear, that the center of action of the seesaw may have changed in time with configuration not precisely similar to today. Summer proxies (Fig. 8d) indicate a prolongation of drier conditions also during the warmer part of the years, suggesting a

persistent Azores high during summer. However, in the central Mediterranean, some records indicate a possible increase of precipitation, possibly as incursions of North Atlantic perturbations, which suggests weakening of the Azores High for some areas, possibly as effect of change is its positions (e.g. Di Rita et al., 2018b). Considering that the position of the ITCZ exerts a control on summer aridity and the temperature in the Mediterranean basin (Eshel et al., 2002; Alpert et al., 2006; Gaetani et al., 2011), a southward shift of the ITCZ may have locally weakened the Azores High, favoring incursion of wetter air from N.

Atlantic. A southward shift of the ITCZ during the 4.2 cal ka BP event is supported by several lines of evidence (e.g. Welc and Marks, 2014; Dixit et al., 2014), and this would have an effect on summer weather over the Mediterranean. Despite evidence are weak it is also reasonable to assume that condition during summer are also important in defining the final "feature" of the 4.2 cal ka BP. Evidence of environmental and climatic deterioration around or coincident with the 4.2 cal ka BP is apparent but chronologically compromised considering a different selection of records (Fig. 9, see also Fig. 1 and Table 1). Reduced

temperature (Fig. 8a) is also consistent with the start of Neoglacial over the Apennines with the first appearance during the Holocene of the Calderone glacier (Zanchetta et al., 2012a). Interestingly, exposure ages in western Alps also indicate a first "Neoglacial advance" at 4.2 ka (Le Roy et al., 2017). This is geographically consistent with lower temperatures during summer, as inferred from chironomids from central Italy (Fig. 9 Samartin et al., 2017). Also in the French side of the Mediterranean Alps, lakes show evidence of high-frequency environmental instabilities during this period. Rapid alternation of drop and rise

of the lake-level are suggested by switch in the benthic/planktonic diatom ratio at Lake Allos (Cartier et al., 2018) while at Lake Petit, more frequent heavy rainfall triggered soil erosion (Fig. 9, Brisset et al., 2013) and ecosystem shift (Cartier et al., 2015). This complex pattern is well explained by oxygen isotope composition performed on diatom cells of Lake Petit that indicates that the period is marked by drier mean conditions (in term of annual lake balance), associated with short-term heavy rainfall (Cartier et al., 2018) rather than just wetter conditions. This is in a good agreement with the other records of the central

Mediterranean. The pollen record from Alimini Piccolo (Fig. 9, Di Rita and Magri, 2009) and from the Gulf of Gaeta (Di Rita et al., 2018b) show a prominent decrease in AP suggesting drier conditions. Drier conditions can be inferred also from a multiproxy record from Qarun Lake in the Faiyum oasis (Egypt, Fig. 1, Fig. 9). At Qarun, the interval between ca. 4 and 4.4 shows an increase in aridity and dust supply, as shown by several proxies (Marks et al., 2018). In north-eastern Africa, there is further evidence of climate change at about 4.2 cal ka BP associated to the collapse of the ancient Old Kingdom in Egypt (Welc

and Marks, 2014, and reference therein). A large data set on condition of aridity in Mediterranean basin is presented by Weiss et al. (this issue). Dust deposition increases in the Eastern Carpathians, as documented by the Moho peat succession (Longman et al., 2017), although an increase of clastic material in a peat succession in a volcanic caldera can be explained also by an increase in local soil erosion. This phase is marked by a change in the d-excess in the ice in the Scărișoara Ice Cave (Carpathian Mountains, Fig. 9, Perșoiu et al., 2017), suggesting a change in the arrival of cyclones sourced from the Mediterranean region.

The dust record in Sidi Ali Lake (Fig. 9) suggests a measurable trans-Saharian aridity event, with increased dust transport at

ca. 4.2 cal ka BP (Zielhofer et al., 2017b). Changes in circulation are also suggested by exotic pollen of Cedar, arriving from North Africa, in some pollen successions of central Italy (e.g. Magri and Parra, 2002). Between ca. 4.4 and 4.0 cal ka BP there is evidence for an increase in storm activity, possibly suggesting an increasing of occasional strong southward incursion of westerlies (Sabatier et al., 2012). This is not in contrast with a trend towards increased flooding in Central Tunisia in this

period correlated with colder period in North Atlantic (Fig. 9, Zielhofer and Faust, 2008).

## 4   Final remarks and trajectories of next researches

The analyses of many records show that between 4.3 and 3.8 cal ka BP climatic and environmental changes occurred in the Mediterranean basin. In many records, it appears evident that an important change in the hydrological regime occurred, with more arid conditions, but locally this evidence is confounded. Cooling can be inferred from different records, but this not

a common feature. Despite contradictory, which is not questioning the evidence of this event, there is the possibility that it is regionally articulated as having a locally different climatic expression. This expression would be also related to different seasonal conditions. From the selected data, the possibility emerges that this event is in reality marked by some oscillations which cannot be resolved in an unequivocal way on the basis of available records. However, regional coverage is still low, even if our record selection is incomplete. The emerging patterns need to be confirmed by future research but well positioned

and well resolved new records can also change our view. Comparing the distribution of the selected records with one of the most important climatic modes like NAO, which impact on Mediterranean climate, in particular in winter, we did not find complete and satisfying matching. We can agree that different working hypotheses investigating the role of potential climatic teleconnections can be inferred from single records or regionally well constrained groups of records, but none seems today convincing. However, this review indicates that many pieces of this complex puzzle are still lacking. Most urgently, new

records at higher resolution with a firm chronological basis are required.

*Code availability.*  TEXT

*Data availability.*  The manuscript is a review and all the data have been collected from previous publication either on open database or asking data directly to the authors. In only one case data have been digitized. In any case, they have been organized in a series of xls files, which can be obtained under request to MB, and GZ.

*Code and data availability.*  TEXT

*Sample availability.*  TEXT





*Author contributions.* MB, GZ and AP conceived the manuscript. MB and GZ organized and written most of the manuscript. In particular MF, RD, II and ER contributed to the construction and writing of section 2.1.1; JRD and RC contributed to the construction and writing section 2.1.2; BJ, M-AS, IC, and FL contributed to the construction and writing section 2.1.3; OP contributed to the construction and writing of the section 2.1.4; FDiR, OP, DM, AM, AMM, and LS contributed to the selection and interpretation of pollen records; LM, AP, MB, 5  FW and CZ contributed to the selection and interpretation of different proxy records along the manuscript. All the co-authors participated in sharing the data and contributed to the scientific discussion.

*Competing interests.* No competing interests are present

*Disclaimer.* TEXT

*Acknowledgements.* MB and GZ are indebted to the University of Pisa and Earth Science Department for the support for organizing the 10  workshop "the 4.2 cal ka BP event". MB and GZ contribution have also been developed within the frame the project:"Climate and alluvial event in Versilia: integration of Geoarcheological, Geomorphological, Geochemical data and numerical simulations" awarded to MB, and funded by the Fondazione Cassa di Risparmio di Lucca. LM and FW were funded by the National Science Centre in Poland (decision no. DEC-2013/09/B/ST10/02040).



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





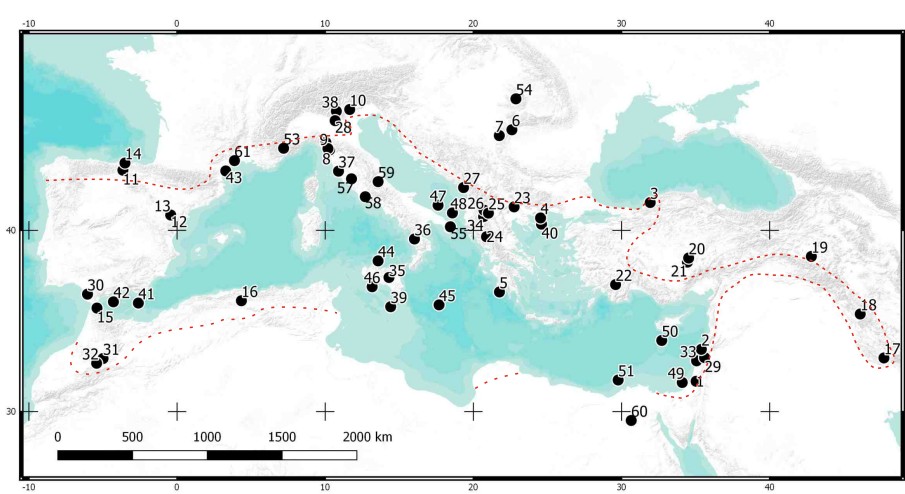

**Figure 1.** Location of selected records discussed in the text. For the numbers and references refers to Table 1. Dotted red line correspond to the limit of growth of Olive tree, taken as roughly indication of Mediterranean climate.

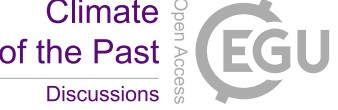



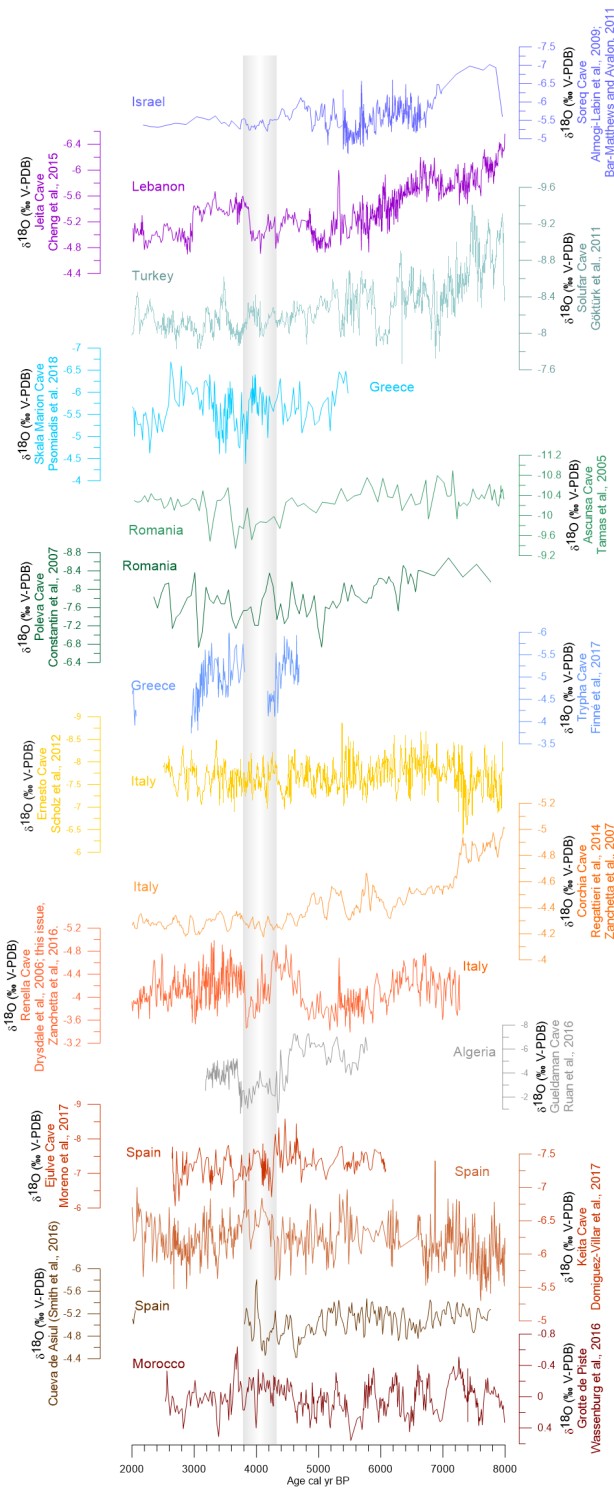

**Figure 2.** Selected speleothem $\delta^{18}$O records. For location refers to Figure 1 and for references to Table 1.



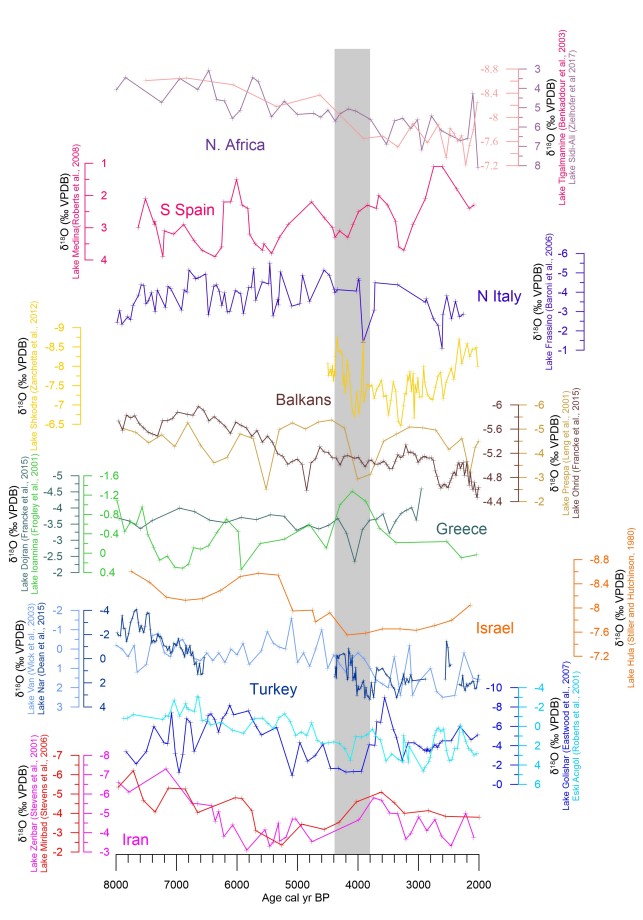

**Figure 3.** Selected lacustrine $\delta^{18}O$ records. For location refers to Figure 1 and for references to Table 1.



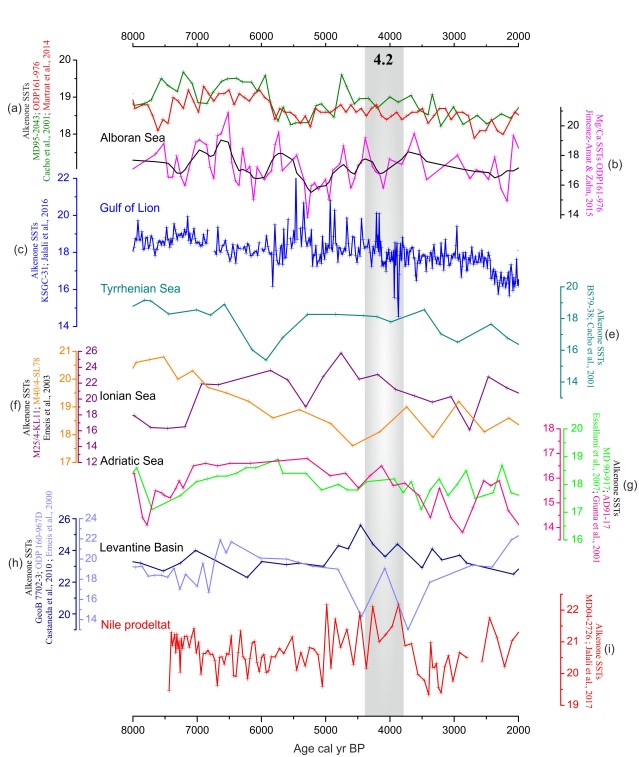

**Figure 4.** Selected Sea Surface Temperature (SST, °C). For location refers to Figure 1 and for references to Table 1.



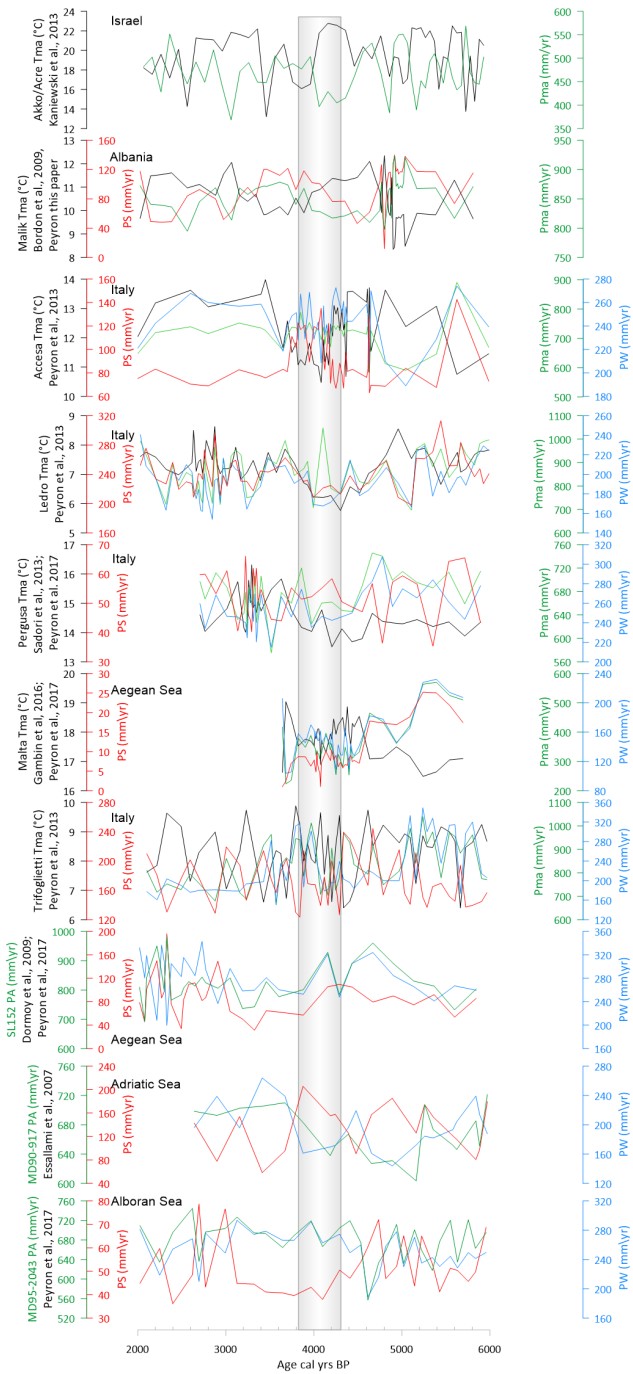

**Figure 5.** Selected temperature (T, °C) and precipitation (P, mm/yr) records obtained from pollen records. Tma: mean annual temperature; Pma: mean annual precipitation; PS: Average summer precipitation; PW: Average winter precipitation. For location refers to Figure 1 and for references to Table 1.





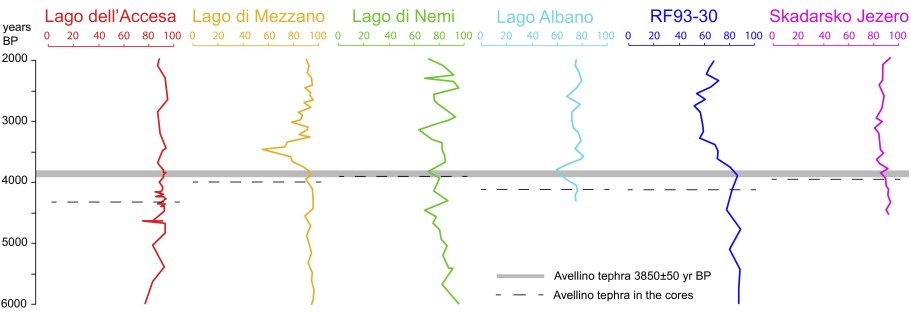

**Figure 6.** Compilation of pollen records containing the Avellino tephra layer (dated at ca. 3.8 cal ka BP, for review see Zanchetta et al., 2018). All the records are plotted with their original age model. Lago dell'Accesa (Drescher-Schneider et al., 2007); Lago di Mezzano (Sadori et al., 2018); Lago Albano and Lago di Nemi (Mercuri et al., 2002); RF93-30 (Mercuri et al., 2012);. For core RF93-30 the correlation with Avellino tephra is not certain (Lowe et al., 2007).





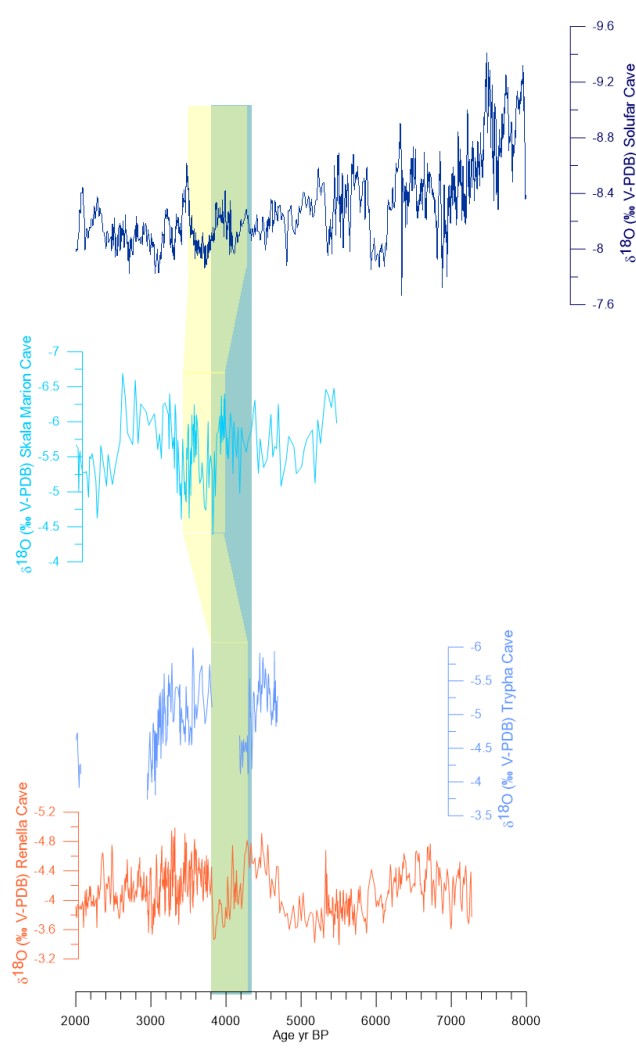

**Figure 7.** Chronologic approach (yellow field) and Climatostratigraphic approach (green field) applied on speleothem records (based on Psomiadis et al., 2018). Renella (after Zanchetta et al., 2016; Drysdale et al. in progress , 2018); Trypha cave (after Finné et al., 2017); Skala Marion cave (after Psomiadis et al., 2018); Solufar cave (after Göktürk et al., 2011).





**Figure 8.** Maps of: a) annual average temperature b) annual average precipitation; c) winter precipitation; d) summer precipitation. See figures 2,3,4, and 5 and Table 1 for data.





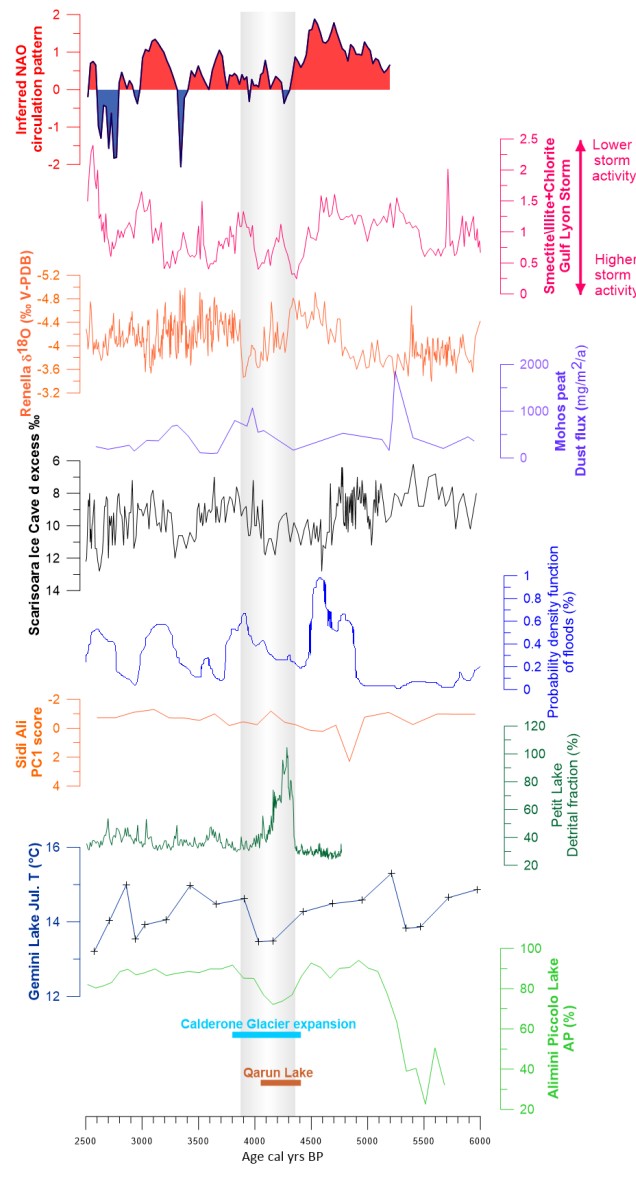

**Figure 9.** Selected additional records to illustrate the general situation over the basin during 4.2 cal ka BP event. NAO-index (Olsen et al., 2011); Storm activity in the Gulf of Lyon (Sabatier et al., 2012); Renella Cave (Zanchetta et al., 2016; Drysdale et al. in progress , 2018); Moho peat dust flux, (Longman et al., 2017); Scărișoara Cave, d-excess in ice cave citepPersoiu2017; Probability flood frequency from N. Tunisa (Zielhofer and Faust, 2008); Sidi Ali dust flux (Zielhofer et al., 2017b); Clastic input at Petit Lake (Brisset et al., 2013); Gemini lake july temperature (Northern Apennine) (Samartin et al., 2017); Alimini Piccolo Arboreal Pollen (AP%) record (Di Rita and Magri, 2009); Calderone Glacier expantions (Zanchetta et al., 2012a); Qarun lake (Marks et al., 2018).

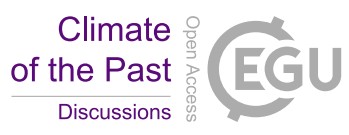

**Table 1.** Sites and proxy records selected in this paper for investigating the 4.2 cal ka BP event. Resolution is reported only for selected proxy and is intended as average during the Holocene.

| n. | Site (archive) | Proxy | Resolution (yr) | Region | Reference |
|---|---|---|---|---|---|
| | *Caves* | | | | |
| 1 | Soreq Cave | $^{18}$O | 16 | Israel | Almogi-Labin et al., 2009; Bar-Matthews and Ayalon, 2011 |
| 2 | Jeita Cave | $\delta^{18}$O | 16 | Lebanon | Cheng et al., 2015 |
| 3 | Solufar Cave | $\delta^{18}$O | 8 | Turkey | Göktürk et al., 2011 |
| 4 | Skala Marion Cave | $\delta^{18}$O | 20 | Greece | Psomiadis et al. 2018 |
| 5 | Trypha Cave | $\delta^{18}$O | 6 | Greece | Finné et al., 2017 |
| 6 | Ascunsa Cave | $\delta^{18}$O | 55 | Romania | Tămaş et al., 2005 |
| 7 | Poleva Cave | $\delta^{18}$O | 76 | Romania | Constantin et al., 2007 |
| 8 | Corchia Cave | $\delta^{18}$O, Mg/Ca | 12 | Italy | Regattieri et al., 2014; Zanchetta et al., 2007 |
| 9 | Renella Cave | $\delta^{18}$O, Mg/Ca | 9 | Italy | Drysdale et al., 2006; this issue, Zanchetta et al., 2016. |
| 10 | Ernesto Cave | $\delta^{18}$O | - | Italy | Scholz et al., 2012 |
| 11 | Keita Cave | $\delta^{18}$O | 10 | Spain | Dominguez-Villar et al., 2017 |
| 12 | Ejulve Cave | $\delta^{18}$O, Mg/Ca | 13 | Spain | Moreno et al., 2017 |
| 13 | Molinos Cave | $\delta^{18}$O, Mg/Ca | 17 | Spain | Muñoz et al., 2015 |
| 14 | Cueva de Asiul | $\delta^{18}$O | 15 | Spain | Smith et al., 2016 |
| 15 | Grotte de Piste | $\delta^{18}$O, Mg/Ca | 15 | Morocco | Wassenburg et al., 2016 |
| 16 | Gueldaman Cave | $\delta^{18}$O | 13 | Algeria | Ruan et al., 2016 |
| | *Lakes* | | | | |
| 17 | Lake Mirabad | $\delta^{18}$O end. calcite | 258 | Iran | Stevens et al., 2006 |
| 18 | Lake Zeribar | $\delta^{18}$O end. calcite | 188 | Iran | Stevens et al., 2001 |
| 19 | Lake Van | $\delta^{18}$O end. calcite | 90 | Turkey | Wick et al., 2003 |
| 20 | Lake Esky Acıgöl | $\delta^{18}$O end. calcite | 88 | Turkey | Roberts et al., 2001 |
| 21 | Lake Nar Gölü | $\delta^{18}$O end. calcite/aragonite | 19 | Turkey | Dean et al., 2015 |
| 22 | Lake Gölhisar | $\delta^{18}$O end. calcite | 97 | Turkey | Jones et al., 2002; Eastwood et al., 2007 |
| 23 | Lake Dojran | $\delta^{18}$O end. calcite | 89 | Republic of Macedonia | Francke et al., 2013 |
| 24 | Ioannina (Lake Pamvotis) | $\delta^{18}$O ostracod | 149 | Greece | Frogley et al., 2001; Roberts et al., 2008 |
| 25 | Lake Prespa | $\delta^{18}$O end. calcite | 157 | Republic of Macedonia | Leng et al., 2010 |
| 26 | Lake Ohrid | $\delta^{18}$O end. calcite | 38 | Republic of Macedonia | Lacey et al., 2015 |
| 27 | Lake Shkodra | $\delta^{18}$O end. calcite | 28 | Albania/Montenegro | Zanchetta et al., 2012 |
| 28 | Lake Frassino | $\delta^{18}$O freshwater mollusk | 136 | Italy | Baroni et al., 2006 |
| 29 | Lake Hula | $\delta^{18}$O end. calcite | 287 | Israel | Stiller and Hutchinson, 1980 |
| 30 | Laguna Medina | $\delta^{18}$O ostracod | 130 | Spain | Roberts et al., 2008 |
| 31 | Lake Sidi-Ali | $\delta^{18}$O ostracod; Dust record | 144 | Morocco | Zielhofer et al., 2017a,b |
| 32 | Lake Tigalmamine | $\delta^{18}$O ostracod | 278 | Morocco | Roberts et al., 2008 |





| n. | Site (archive) | Proxy | Resolution (yr) | Region | Reference |
|---|---|---|---|---|---|
| | *Pollen* | | | | |
| 33 | Akko | Pollen (P, T) | 85 | Israel | Kaniewski et al., 2013 |
| 34 | Maliq | Pollen (P, T) | 87 | Albania | Bordon et al., 2009, Peyron this paper |
| 35 | Pergusa | Pollen (P, T) | 154 | Italy | Sadori et al., 2013; Peyron et al. 2017 |
| 36 | Trifoglietti | Pollen (P, T) | 73 | Italy | Peyron et al., 2013 |
| 37 | Accesa | Pollen (P, T) | 97 | Italy | Peyron et al., 2013 |
| 38 | Ledro | Pollen (P, T) | 66 | Italy | Peyron et al., 2013 |
| 39 | Butmarrad | Pollen (P, T) | 138 | Aegean Sea | Gambin et al, 2016; Peyron et al., 2017 |
| 40 | SL152 | Pollen (P) | 76 | Aegean Sea | Dormoy et al., 2009; Peyron et al., 2017 |
| 41 | MD95-2043 | Pollen (P) | 106 | Alboran Sea | Peyron et al., 2017 |
| 42 | ODP-976 | Pollen (P) | 129 | Alboran Sea | Dormoy et al., 2009; Peyron et al., 2017 |
| | *Marine* | | | | |
| 41 | MD95-2043 | Alkenone SST | 110 | Alboran Sea | Cacho et al., 2001 |
| 42 | ODP-976 | Alkenone SST, Mg/Ca SST | 34 | Alboran Sea | Martrat et al., 2014; Jimenez-Amat and Zahn, 2015 |
| 43 | KSGC-31 | Alkenone SST | 15 | Gulf of Lion | Jalali et al., 2016 |
| 44 | BS79-38 | Alkenone SST | 59 | Tyrrhenian Sea | Cacho et al., 2001 |
| 45 | M25/4-KL11 | Alkenone SST | 260 | Ionian Sea | Emeis et al., 2000 |
| 46 | M40/4-SL78 | Alkenone SST, | 160 | Ionian Sea | Emeis et al., 2000 |
| 47 | MD90-917 | Alkenone SST, | 40 | Adriatic Sea | Essallami et al., 2007 |
| 48 | AD91-17 | Alkenone SST | 190 | Adriatic Sea | Giunta et al., 2001 |
| 49 | GeoB 7702-3 | Alkenono SST | 210 | Levantine Basin | Castaneda et al., 2010 |
| 50 | ODP 160-967D | Alkenone SST | 94 | Levantine Basin | Emeis et al., 2000 |
| 51 | MD04-2726 | Alkenone SST | 57 | Nile prodelta | Jalali et al., 2017 |
| | *Other records* | | | | |
| 52 | Mohos Bog | Dust record | | Romania | Longman et al., 2017 |
| 53 | Lac Petit | Detrital fraction | | France | Brisset et al., 2013 |
| 54 | Scărişoara Cave | D-excess in ice | | Romania | Perşoiu et al., 2017 |
| 55 | Alimini Piccolo | Pollen | | Italy | Di Rita and Magri, 2009 |
| 56 | Gemini Lake | July T | | Italy | Samartin et al., 2015 |
| 57 | Mezzano Lake | Pollen | | Italy | Sadori, 2018 |
| 58 | Albano and Nemi lakes | Pollen | | Italy | Mercuri et al., 2002 |
| 59 | Calderone Glacier | Glacier record | | Italy | Zanchetta et al., 2012b |
| 60 | Qarun Lake | Lake level | | Egypt | Marks et al., 2018 |
| 61 | BP-06 | Storminess record | | France | Sabatier et al., 2012 |
| 62 | Tunisia | Flood record | | Tunis | Zielhofer and Faust, 2008 |

Average resolution during the Holocene