# Peer review of "The 4.2 ka BP Event in the Mediterranean Region: an overview"

_Climate of the Past, 2018_

## Referee Comment (RC1) · Anonymous Referee #1 · 20 Nov 2018

The manuscript presented by the authors seems to me of great relevance, because of the paleoenvironmental and archaeological interest of the subject and the discourse and arguments that they develop. Honestly, I consider the title chosen by the authors in their paper of great honesty, since they do not try to make us suppose that it is a synthesis of the issue but simply a summary or overview. My most sincere congratulations to the authors for this magnificent contribution. It is true, however, that I miss some kind of archaeological implication of the data presented in this manuscript; although I also understand that it is not your goal, but it would have been a great contribution.

The manuscript hosts a very adequate and convincing speech, its organization is perfect and the volume of data handled is impressive. The results and discussion are wonderful and very well done. Once again, my most sincere congratulations to the

authors.

Suggestions and minor corrections: + Page 2 Line 25. Include these two references: Blanco-González, A., Lillios, K.T., López-Sáez, J.A. & Drake, B.L. (2018). Cultural, demographic and environmental dynamics of the Copper and Early Bronze Age in Iberia (3300-1500 BC): towards an interregional multiproxy comparison at the time of the 4.2 ky BP event. Journal of World Prehistory, 31: 1-79. Lillios, K.T., Blanco González, A., Drake, B.L. & López-Sáez, J.A. (2016). Mid-Late Holocene climate, demography, and cultural dynamics in Iberia: a multi-proxy approach. Quaternary Science Reviews, 135: 138-153. + Page 3 Line 12. The authors are right in their statements. However, I believe that they could be much more precise in explaining the chosen records. For instance, on line 18 they themselves speak of "optimal" conditions. What are these conditions? The authors should make this issue clear since it is probably the most important in all its argumentation. + It is evident, and quite logical, that the authors have made a very detailed selection of the records used and discussed in this paper. However, after reading the methods section several times, I still do not have clear concepts and reasons for such choice. For example, only 5 records have been selected in the Iberian Peninsula, and none of them correspond to pollen studies. Why? In Iberian territory there are numerous pollen records at high resolution that could yield information of great value to treat the 4.2 ka cal BP event. I understand perfectly that it is impossible to use all the available information and that authors have to select conveniently; but at least, the authors should provide a convincing explanation of the choice of records. In reference to the previous point, a very great possibility that the authors could have used, would have been to select those records that have several proxies, for example, pollen and ostracods / oxygen isotopes. This could have been the case, and I am speaking by heart, of some of the records cited in Table 1, such as Prespa, Ohrid, Medina Lagoon, Tigalmamine, Sidi-Ali, Lac Petit, etc. I do not understand why there are several proxies for these records, the authors, as indicated in Table 1, have only taken into account one for each record. The discussion is sincerely incredible, very good in its speech and argumentation. Points 3.1 to 3.5 combine all the results with sufficiency and clairvoyance.

Please also note the supplement to this comment:
https://www.clim-past-discuss.net/cp-2018-147/cp-2018-147-RC1-supplement.pdf

———————————————————

---

## Referee Comment (RC2) · Anonymous Referee #2 · 30 Nov 2018

The manuscript by Bini et al. presents a thorough review of the 4.2 "event" in the Mediterranean, based on 62 records from the area. The paper contains a long list of coauthors who are all experts in their various fields/proxy types. The paper is well written and nicely illustrated, and the arguments set forward by the authors are both convincing and pertinent. Their review underscores the complexity of the event. A Mediterranean-wide update of the "4.2 event" is long overdue. In sum, I fully support the publication of the paper in Climate of the Past, with some minor revisions.

Some minor suggestions Page 3, Lines 5-6: "In this paper we use the term "4.2 cal ka BP event" to indicate a period of time between ca. 4.3 and 3.8 cal ka BP (close to the definition of Weiss (2015, 2016)." This is a good point. I personally think that the term "event" is not well adapted to describe what is a fairly protracted period of climate

change.

Page 3: The authors transparently outline the limitations and challenges of the records included in their review.

Page 6, line 7 : missing lake name after and. . .

Page 12, line 7: As demonstrated in Marriner et al. (2012) QSR, the position and intensity of the ITCZ is also affected by ENSO. It might be worth considering ENSO variability here. The record from Zhua et al. (2017, PNAS) "Holocene ENSO-related cyclic storms recorded by magnetic minerals in speleothems of central China" shows clear evidence for increased ENSO variability around 4.2 which could have implications for the Mediterranean?

Page 13, line 4: See also: - Kaniewski, D., Marriner, N., Morhange, C., Faivre, S., Otto, T., Van Campo, E. (2016). Solar pacing of storm surges, coastal flooding and agricultural losses in the Central Mediterranean. Scientific Reports, 6, 25197. - Marriner, N., Kaniewski, D., Morhange, C., Flaux, F., Giaime, M., Vacchi, M., Goff, J. (2017). Tsunamis in the geological record: making waves with a cautionary tale from the Mediterranean. Science Advances, 3, 10, e1700485.

There is no real discussion of the possible cultural implications of the 4.2 climate event on Mediterranean societies. This could be useful.

Figure 2: Many of the records show evidence for insolation-based aridification. I suggest that this authors plot an insolation curve for Mediterranean latitudes to show this.

---

## Referee Comment (RC3) · Anonymous Referee #3 · 18 Dec 2018

Beni et al., by far, laid out the state of the knowledge on 4.2 event in the Mediterranean region. The review is thorough and underscores most of the issues associated with 4.2 climate event and discussed the complexities caused by chronology, record expansion, and sensitivity of chosen proxies to the climate variability. One of the significance of this contribution is discussing the shortcomings of the available paleoclimate records from the region with respect to tracing abrupt climate variabilities such as 4.2 and related forcing mechanism. I found the manuscript well organized with adequate discussion and convincing conclusion. It is perfectly fit for the publication in Climate of the Past Discussion, with some minor revisions.

ć Page 3, Line 19: The authors stated that from paleoclimate archive they chose "the most powerful proxy" to reconstruct climate. Please elaborate more on how you eval-
uated the efficiency of the proxies in reflecting regional climate. Although in previous paragraphs the authors clearly stated that the selection of the records as well as proxies was, to some extent, subjective but it would be great if they can provide some mathematical methods, such as using probability function, to highlight the records/proxies with highest probabilities in showing 4.2 event. They can test this on one or two records just to show the validity of their selection. the results could be presented in supplementary materials. aÅć Page 10, Line 8: The author chose paleoclimate records from Lake Zeribar and Lake Mirabad at the eastern end of their climate records. These two lakes are located in Zagros Mountains with very complex and poorly understood climate condition. Both records are lacking high resolution and optimal chronology as mentioned by the authors. It is highly recommended to replace these two records with the pollen record from Lake Maharlou in Zagros (Djamali et al, 2009) and multi-proxy record from Neor Lake, NW Iran (Sharifi et al, 2015). These records have optimal resolution with robust chronology and both clearly captured the 4.2 event. REFERENCES: Djamali et al. 2009: Vegetation history of the SE section of the Zagros Mountains during the last five millennia; a pollen record from the Maharlou Lake, Fars Province, Iran. Veget Hist Archaeobot (2009) 18:123–136, DOI 10.1007/s00334-008-0178-2. Sharifi et al., 2015: Abrupt climate variability since the last deglaciation based on a high-resolution, multi-proxy peat record from NW Iran: The hand that rocked the Cradle of Civilization? Quaternary Science Reviews, Volume 123, pp. 215-230. ć Page 11, Lines 31 and 33: Please include the so-called Figure 10 in the manuscript. aAć Page 12, Lines 7-11: This is a great point and correlates well with the southward shift of the mid-latitude westerly jet (MLWJ). It might be worth considering the interplay of ITCZ and MLWJ and its effect on precipitation over the Mediterranean. Brayshaw et al. (2010) studied the changes in winter storm track over the North Atlantic and the Mediterranean during the Holocene and Nagashima et al. (2011) showed the changes in the westerly jet path during the last glacial period. The TraCE simulation conducted by Sharifi et al. (2018) revealed an equatorward shift in the position of westerly jet throughout the Holocene with an abrupt shift centered at 4.2 ka B.P. REFERENCES: Bravshaw, D.J., Hoskins,
B., Black, E., 2010. Some physical drivers of changes in the winter storm tracks over the North Atlantic and Mediterranean during the Holocene. Philos. Trans. R. Soc.A, Math. Phys. Eng. Sci.368, 5185–5223. http://dx.doi.org/10.1098/rsta.2010.0180. Nagashima, K., Tada, R., Tani, A., Sun, Y., Isozaki, Y., Toyoda, S., Hasegawa, H., 2011. Millennial-scale oscillations of the westerly jet path during the last glacial period. J. Asian Earth Sci.40, 1214–1220. http://dx.doi.org/10.1016/j.jseaes.2010.08.010. Sharifi, A.; Murphy, L. N.; Pourmand, A.; Clement, A. C.; Canuel, E. A.; Naderi Beni, A.; Lahijani, H. A. K. and Ahmady-Birgani, H., 2018:Early Holocene Greening of the Afro-Asian Dust Belt Changed Sources of Mineral Dust in West Asia. Earth and Planetary Science Letters, Volume 481, pp.30-40, DOI 10.1016/j.epsl.2017.10.001.

---

## Author Comment (AC2) · 23 Jan 2019

Because some requests are similar, we replay in a unique file for all referees.

Reply to the Anonymous Referee #1

We thank the Referee #1 for the useful comments and suggestions. Below we detail our reply point by point.

The manuscript presented by the authors seems to me of great relevance, because of the paleoenvironmental and archaeological interest of the subject and the discourse and arguments that they develop. Honestly, I consider the title chosen by the authors in their paper of great honesty, since they do not try to make us suppose that it is a synthesis of the issue but simply a summary or overview. My most sincere congratulations

to the authors for this magnificent contribution. The manuscript hosts a very adequate and convincing speech, its organization is perfect and the volume of data handled is impressive. The results and discussion are wonderful and very well done. Once again, my most sincere congratulations to the authors.

We thank the ref#1 for the general positive comment.

Suggestions and minor corrections: Page 2 Line 25. Include these two references: Blanco-González, A., Lillios, K.T., López-Sáez, J.A. & Drake, B.L. (2018). Cultural, demographic and environmental dynamics of the Copper and Early Bronze Age in Iberia (3300-1500 BC): towards an interregional multiproxy comparison at the time of the 4.2 ky BP event. Journal of World Prehistory, 31: 1-79. Lillios, K.T., Blanco González, A., Drake, B.L. & López-Sáez, J.A. (2016). Mid-Late Holocene climate, demography, and cultural dynamics in Iberia: a multi-proxy approach. Quaternary Science Reviews, 135: 138-153.

We thank the reviewer for the suggestions. These are very interesting papers and we decided to quote them not where suggested but later when we discussed the Methods and Blanco-Gonzalez et al is also quoted later in the discussion.

Page 3 Line 12. The authors are right in their statements. However, I believe that they could be much more precise in explaining the chosen records. For instance, on line 18 they themselves speak of "optimal" conditions. What are these conditions? The authors should make this issue clear since it is probably the most important in all its argumentation.

This is a similar request of the referee 3. We have added some sentences in the Methods to remark the criteria used in the selection. However, we think most of the explanations are already included in the old version at page. 3-4 starting line 19. The values of the selected proxies are again explained in all sections referring to specific proxies used. We have also clearly stated that we are aware of the potential limitation of our selection. At the sentence (line 19 pag. 3 of the old version): "For each

archive, we have selected the most powerful (or considered as such) proxy to reconstruct climate", we substitute the following sentence with: "Among a copious number of data showing, even if with different expression, the 4.2 ka event and its impact in the Mediterranean basin (e.g. Magny et al., 2009; Margaritelli et al. 2016; Blanco-González et al., 2018) , we have decided to select only the proxies that can give, in our opinion, tighter information on the hydrological variability like oxygen isotope composition of continental carbonates (e.g. Roberts et al., 2010) and on the temperature conditions at regional scale, as reconstructed by pollen data and marine proxies (Jalali et al., 2016; Kaniewski et al., 2018)."

It is evident, and quite logical, that the authors have made a very detailed selection of the records used and discussed in this paper. However, after reading the methods section several times, I still do not have clear concepts and reasons for such choice. For example, only 5 records have been selected in the Iberian Peninsula, and none of them correspond to pollen studies. Why? In Iberian territory there are numerous pollen records at high resolution that could yield information of great value to treat the 4.2 ka cal BP event. I understand perfectly that it is impossible to use all the available information and that authors have to select conveniently; but at least, the authors should provide a convincing explanation of the choice of records. In reference to the previous point, a very great possibility that the authors could have used, would have been to select those records that have several proxies, for example, pollen and ostracods / oxygen isotopes. This could have been the case, and I am speaking by heart, of some of the records cited in Table 1, such as Prespa, Ohrid, Medina Lagoon, Tigalmamine, Sidi-Ali, Lac Petit, etc. I do not understand why there are several proxies for these records, the authors, as indicated in Table 1, have only taken into account one for each record.

We agree with the referee that there are a huge amount of proxies, which are not considered in this manuscript. We agree that there are many other records that can give important insight on the matter. However, the huge amount of proxies can also be

difficult to manage and the kind of information that can be obtained can be very different (edaphic condition, primary productivity, local phenomena, human impact ect.) and not all these information are really climatic. We think we have clarified that we focus on proxies with high hydrological component (oxygen stable isotopes on lacustrine carbonate and speleothems) and are able to give quantitative reconstruction on P and T (using pollen and marine records) and we focus only on these records. So, we eliminate from our discussion the other records/proxies. To give an idea of the different sources of data and information we produced fig. 9. It wants be (as stated) just an example of different potential information in other records, and any can argue that some records are missing. We are aware for this and also the reader must be. I think the referee should be aware that in the special issues there are many other contributions and that some specifically focus on the Iberian Peninsula with a multiproxy approach, complementing our approach.

The discussion is sincerely incredible, very good in its speech and argumentation. Points 3.1 to 3.4 combine all the results with sufficiency and clairvoyance.

Thanks for these final comments

Replay to the Anonymous Referee #2

We thank the Referee #2 for the useful comments and suggestions. Below we detail our reply point by point.

The manuscript by Bini et al. presents a thorough review of the 4.2 "event" in the Mediterranean, based on 62 records from the area. The paper contains a long list of coauthors who are all experts in their various fields/proxy types. The paper is well written and nicely illustrated, and the arguments set forward by the authors are both convincing and pertinent. Their review underscores the complexity of the event. A Mediterranean-wide update of the "4.2 event" is long overdue. In sum, I fully support the publication of the paper in Climate of the Past, with some minor revisions.

We thank the ref#2 for the general positive comment.

Some minor suggestions

Page 3, Lines 5-6: "In this paper we use the term "4.2 cal ka BP event" to indicate a period of time between ca. 4.3 and 3.8 cal ka BP (close to the definition of Weiss (2015, 2016)." This is a good point. I personally think that the term "event" is not well adapted to describe what is a fairly protracted period of climate change.

We agree, this is a popular way to indicate a climatic event (see for instance also 8.2 event or others), even if they often represent a period of different duration.

Page 3: The authors transparently outline the limitations and challenges of the records included in their review.

Yes, we think this is a very fundamental point to be discussed and the reader must be aware of the limitation of our conclusions.

Page 6, line 7 : missing lake name after and:

Because the sentence was not probably so clear, we have changed it a little.

Now it is: "Some of the records reported in Roberts et al. (2008) and some new records that were too short, or with too low resolution or poor chronologic accuracy (e.g. Lake Pergusa, Zanchetta et al., 2007b; Valle di Castiglione, Zanchetta et al., 1999; Lake Yammoûneh, Develle et al., 2010), have been excluded from this review.

Page 12, line 7: As demonstrated in Marriner et al. (2012) QSR, the position and intensity of the ITCZ is also affected by ENSO. It might be worth considering ENSO variability here. The record from Zhua et al. (2017, PNAS) "Holocene ENSO-related cyclic storms recorded by magnetic minerals in speleothems of central China" shows clear evidence for increased ENSO variability around 4.2 which could have implications for the Mediterranean?

Interesting suggestion. We have inserted in the text: "Marriner et al. (2012) have

showed a decrease in Nile delta flood-driven accretion between ca. 4.4 and 4.1 cal ka BP in response to weakening of ITCZ, related to changes in El Niño Southern Oscillation type (ENSO) variability. This may indicate an indirect influence of ENSO variability on climate of the Mediterranean during 4.2 ka event"

Page 13, line 4: See also: - Kaniewski, D., Marriner, N., Morhange, C., Faivre, S., Otto, T., Van Campo, E. (2016). Solar pacing of storm surges, coastal flooding and agricultural losses in the Central Mediterranean. Scientific Reports, 6, 25197. - Marriner, N., Kaniewski, D., Morhange, C., Flaux, F., Giaime, M., Vacchi, M., Goff, J. (2017). Tsunamis in the geological record: making waves with a cautionary tale from the Mediterranean. Science Advances, 3, 10, e1700485.

We have inserted in the text the following sentence (which refers to fig. 9): "Between ca. 4.4 and 4.0 cal ka BP there is evidence for an increase in storm activity as documented by several records in the central Mediterranean (Sabatier et al., 2012; Kaniewski et al., 2016, Marriner et al., 2017), possibly suggesting an increasing of occasional strong southward incursion of westerlies".

There is no real discussion of the possible cultural implications of the 4.2 climate event on Mediterranean societies. This could be useful.

Yes, indeed. This is a clear choice of the 4.2 workshop and for this special issue. We want to understand first the timing, progression, intensity and duration of the 4.2 event. and we have also intended to present a "climate-only" review that could be further used by archaeologists/historians in their assessment of the cultural implication of past climate changes. A discussion of the cultural implications of the 4.2 ka event would have had to be as comprehensive as the review of the climatic conditions, thus difficult to include in our paper.

Figure 2: Many of the records show evidence for insolation-based aridification. I suggest that this authors plot an insolation curve for Mediterranean latitudes to show this. Yes, for some records we agree, it is true. However, in our opinion to put insolation in

figure 2 is probably confounding (why not in the others? Several figures are already very dense of data). This dependence is not completely visible in other records and inserting insolation in the set of figures comprising different proxies is not probably very useful. Further, the various proxies record may represent different seasons, and as such several insolation curves would be necessary (e.g., JJA, DJF etc). While useful, this would steer the discussion away from the main topic of our paper. However, it would be useful for the readers to have a reference curve for insolation and we have inserted it in fig. 9.

Replay to the Anonymous Referee #3

We thank the Referee #3 for the useful comments and suggestions. Below we detail our reply point by point.

Bini et al., by far, laid out the state of the knowledge on 4.2 event in the Mediterranean region. The review is thorough and underscores most of the issues associated with 4.2 climate event and discussed the complexities caused by chronology, record expansion, and sensitivity of chosen proxies to the climate variability. One of the significance of this contribution is discussing the shortcomings of the available paleoclimate records from the region with respect to tracing abrupt climate variabilities such as 4.2 and related forcing mechanism. I found the manuscript well organized with adequate discussion and convincing conclusion. It is perfectly fit for the publication in Climate of the Past Discussion, with some minor revisions. We thank the ref#3 for the general positive comment.

Page 3, Line 19: The authors stated that from paleoclimate archive they chose "the ʹ most powerful proxy" to reconstruct climate. Please elaborate more on how you evaluated the efficiency of the proxies in reflecting regional climate. We have added some sentences in the Methods to remark the criteria used in the selection. However, we think most of the explanations are already included in the old version at pag. 3-4 starting line 19. The values of the selected proxies is again explained in all sections

referring to specific proxies used. We have also clearly stated that we are aware of the potential limitation of our selection.

At the sentence (line 19 pag. 3 of the old version): "For each archive, we have selected the most powerful (or considered as such) proxy to reconstruct climate", we substitute the following sentence: "Among a copious number of data showing, even if with different expression, the 4.2 ka event and its impact in the Mediterranean basin (e.g. Magny et al., 2009; Margaritelli et al. 2016; Blanco-González et al., 2018) , we have decided to select only the proxies that can give, in our opinion, tighter information on the hydrological variability like oxygen isotope composition of continental carbonates (e.g. Roberts et al., 2010) and on the temperature conditions at regional scale, as reconstructed by pollen data and marine proxies (Jalali et al., 2016; Kaniewski et al., 2018)."

-Although in previous paragraphs the authors clearly stated that the selection of the records as well as proxies was, to some extent, subjective but it would be great if they can provide some mathematical methods, such as using probability function, to highlight the records/proxies with highest probabilities in showing 4.2 event. They can test this on one or two records just to show the validity of their selection. The results could be presented in supplementary materials. Yes, this was one of the possible options at the beginning of the discussion when we decide how to organize the data for the paper, but considering the number of records and their different quality this would have generated a different kind of problem to be solved and the final meaning of the manuscript would be substantially different. However, this would be of great interest for a different work for the future. Indeed, this would include a more restricted selection of records having similar, high chronological resolution and probably including different proxies. A nice example of a simple mathematical treatment of the data is reported in Isola et al 2018 (this issue), which can be expanded in the near future. Page 10, Line 8: The author chose paleoclimate records from Lake ʹ Zeribar and Lake Mirabad at the eastern end of their climate records. These two lakes are located in Zagros Mountains with very complex and poorly understood climate condition. Both

records are lacking high resolution and optimal chronology as mentioned by the authors. It is highly recommended to replace these two records with the pollen record from Lake Maharlou in Zagros (Djamali et al, 2009) and multi-proxy record from Neor Lake, NW Iran (Sharifi et al, 2015). These records have optimal resolution with robust chronology and both clearly captured the 4.2 event. REFERENCES: Djamali et al, 2009: Vegetation history of the SE section of the Zagros Mountains during the last five millennia; a pollen record from the Maharlou Lake, Fars Province, Iran. Veget Hist Archaeobot (2009) 18:123–136, DOI 10.1007/s00334-008-0178-2. Sharifi et al., 2015: Abrupt climate variability since the last deglaciation based on a high-resolution, multi-proxy peat record from NW Iran: The hand that rocked the Cradle of Civilization? Quaternary Science Reviews, Volume 123, pp. 215-230. We thank for these suggestions. However, it is clear from our contribution that we selected $\delta18O$ on lacustrine carbonate and T and P reconstructed from pollen records. Others records are not included in our discussion. The record selected for figure 9 are indicative of different possible information, which can be obtained. We note that Djmali et al., 2009 paper deals with pollen without any precise reconstruction in T or P. Moreover, the 4.2 ka BP even is not so clear in this record. We agree that Sharifi et al. 2015 is better resolved and chronologically robust. It has been quoted later in the discussion. However, for comparison with the other records lacks oxygen isotope data from lacustrine carbonates. Page 11, Lines 31 and ′ 33: Please include the so-called Figure 10 in the manuscript. Yes, right. Sorry for the terrible inadvertency Page 12, Lines 7- 11: This is a great point and correlates well with the southward shift of the mid-latitude westerly jet (MLWJ). It might be worth considering the interplay of ITCZ and MLWJ and its effect on precipitation over the Mediterranean. Brayshaw et al. (2010) studied the changes in winter storm track over the North Atlantic and the Mediterranean during the Holocene and Nagashima et al. (2011) showed the changes in the westerly jet path during the last glacial period. The TraCE simulation conducted by Sharifi et al. (2018) revealed an equatorward shift in the position of westerly jet throughout the Holocene with an abrupt shift centered at 4.2 ka B.P. REFERENCES: Brayshaw, D.J., Hoskins,

C2 CPD Interactive comment Printer-friendly version Discussion paper B., Black, E., 2010. Some physical drivers of changes in the winter storm tracks over the North Atlantic and Mediterranean during the Holocene. Philos. Trans. R. Soc.A, Math. Phys. Eng. Sci.368, 5185–5223. http://dx.doi.org/10.1098/rsta.2010.0180. Nagashima, K., Tada, R., Tani, A., Sun, Y., Isozaki, Y., Toyoda, S., Hasegawa, H., 2011. Millennial-scale oscillations of the westerly jet path during the last glacial period. J. Asian Earth Sci.40, 1214–1220. http://dx.doi.org/10.1016/j.jseaes.2010.08.010. Sharifi, A.; Murphy, L. N.; Pourmand, A.; Clement, A. C.; Canuel, E. A.; Naderi Beni, A.; Lahijani, H. A. K. and Ahmady-Birgani, H., 2018:Early Holocene Greening of the AfroAsian Dust Belt Changed Sources of Mineral Dust in West Asia. Earth and Planetary Science Letters, Volume 481, pp.30-40, DOI 10.1016/j.epsl.2017.10.001.

We thank the reviewer for this suggestion. Indeed, this is a point to focus in the future. We added some sentences in the discussion: "Brayshaw et al. (2010) discussed the influence of the position of the Mid-latitude westerly jet (MLWJ) over the winter precipitation in the Mediterranean. Their modelling indicate a southward shift of the MLWJ during the second part of the Holocene with related changes in cyclogenesis over Mediterranean. The importance in the shift of the position of the MLWJ is also documented in dust proxy records from Middle East and East Asia (e.g. Nagashima et al., 2011; Sharafi et al., 2015, 2018). According to Sharafi et al. (2018), evidence from the dust record from Neor peat mire in Iran and climate modelling show that at ca. 4.2 ka there is a migration of the main axis of the MLWJ towards the equator allowing the transport of higher fluxes of dust from West Asia as well as from the NE African. This indicates a complex but possibly correlated interplay between the ITCZ, MLWJ and Mediterranean precipitation."

Additional sentence inserted

Considering the submission of for the special issue of the manuscript PerÈŹoiu et al. we have inserted a further sentence at pag. 12 line 2.

"It is interesting to note that PerÈŹoiu et al. (2018) suggested that part of the drier and cold condition over Mediterranean during the 4.2 ka event was caused by the strengthening and expansion of the Siberian High, which effectively blocked the moisture-carrying westerlies and enhanced outbreaks of cold and dry winds."

Considering the submission for the special issue of the manuscript (now accepted) Isola et al. 2018 we have inserted a further sentence at pag. 11 line 29.

"This is further confirmed by new speleothem data (stable isotopes and trace elements) reported by Isola et al. (2018), from Apuan Alps in central Italy."

Considering the submission for the special issue of the manuscript Catala et al. 2018 we have inserted a further sentence at pag. 8 line 28.

[revised manuscript text omitted]

Updating affiliations:

Fabrizio Lirer: Istituto di Scienze Marine (ISMAR)-CNR Napoli, Italy

Please also note the supplement to this comment:
https://www.clim-past-discuss.net/cp-2018-147/cp-2018-147-AC2-supplement.pdf

---

## Author Response (AR1)

Dear Editor,

we have now completed our revision of the manuscript. We thank you and the reviewers for the positive comments which helped us to improve the final quality of the manuscript.

We uploaded the final version of the manuscript

Best Regards

Monica Bini